# Amyloid-Beta Peptides Trigger Premature Functional and Gene Expression Alterations in Human-Induced Neurons

**DOI:** 10.3390/biomedicines11092564

**Published:** 2023-09-18

**Authors:** Ana Raquel Melo de Farias, Alexandre Pelletier, Lukas Cruz Carvalho Iohan, Orthis Saha, Amélie Bonnefond, Philippe Amouyel, Fabien Delahaye, Jean-Charles Lambert, Marcos R. Costa

**Affiliations:** 1Université de Lille, Inserm, CHU Lille, Institut Pasteur de Lille, U1167-RID-AGE Facteurs de Risque et Déterminants Moléculaires des Maladies Liées au Vieillissement, DISTALZ, 1 rue du Professeur Calmette, 59019 Lille, France; ana-raquel.melo-de-farias@pasteur-lille.fr (A.R.M.d.F.); orthis@gmail.com (O.S.); philippe.amouyel@pasteur-lille.fr (P.A.); jean-charles.lambert@pasteur-lille.fr (J.-C.L.); 2Brain Institute, Federal University of Rio Grande do Norte, Campus Universitário Lagoa Nova, Av. Senador Salgado Filho, 3000, Natal 59078-970, Brazil; 3Université de Lille, Inserm, CNRS, CHU Lille, Institut Pasteur de Lille, U1283-UMR 8199 EGID, Pôle Recherche, 1 Place de Verdun, CEDEX, 59045 Lille, France; adpelle1@bu.edu (A.P.); amelie.bonnefond@cnrs.fr (A.B.); fabien.delahaye@pasteur-lille.fr (F.D.); 4Bioinformatics Multidisciplinary Environment, Federal University of Rio Grande do Norte, Natal 59078-970, Brazil; lukas@neuro.ufrn.br

**Keywords:** amyloid-beta peptides, human-induced neurons, single-nucleus RNA sequencing, calcium imaging, inter-cellular communication

## Abstract

Alzheimer’s disease (AD) is the most prevalent cause of dementia in the elderly, characterized by the presence of amyloid-beta (Aβ) plaques, neurofibrillary tangles, neuroinflammation, synapse loss and neurodegeneration in the brain. The amyloid cascade hypothesis postulates that deposition of Aβ peptides is the causative agent of AD pathology, but we still lack comprehensive understanding of the molecular mechanisms connecting Aβ peptides to neuronal dysfunctions in AD. In this work, we investigate the early effects of Aβ peptide accumulation on the functional properties and gene expression profiles of human-induced neurons (hiNs). We show that hiNs acutely exposed to low concentrations of both cell-secreted Aβ peptides or synthetic Aβ_1–42_ exhibit alterations in the frequency of calcium transients suggestive of increased neuronal excitability. Using single-cell RNA sequencing, we also show that cell-secreted Aβ up-regulates the expression of several synapse-related genes and down-regulates the expression of genes associated with metabolic stress mainly in glutamatergic neurons and, to a lesser degree, in GABAergic neurons and astrocytes. These neuronal alterations correlate with activation of the SEMA5, EPHA and NECTIN signaling pathways, which are important regulators of synaptic plasticity. Altogether, our findings indicate that slight elevations in Aβ concentrations are sufficient to elicit transcriptional changes in human neurons, which can contribute to early alterations in neural network activity.

## 1. Introduction

Alzheimer’s disease (AD) is a neurodegenerative disorder associated with severe cognitive impairment in which memory loss is one of the most predominant features. Worldwide, AD accounts for 60 to 80 percent of dementia cases and represents an increasing burden for ageing populations [1]. Analyses of post-mortem human brains reveal two key features of AD, namely the presence of amyloid plaques—extra-cellular accumulations of amyloid-β (Aβ) peptides, which are derived from the proteolytic processing of the amyloid precursor protein (APP); and neurofibrillary tangles—intra-cellular accumulations of the microtubule-associated protein tau. The identification of familial, AD-linked mutations in the genes for amyloid-β precursor protein (APP) and presenilin (PS1 and PS2) associated with deregulation of Aβ peptide production suggests that APP metabolism is at the heart of the disease process through the statement of the amyloid cascade hypothesis [2]. Remarkably, recent treatments developed from this hypothesis and based on immunotherapies against Aβ peptides have shown significant but limited clinical effects in slowing down disease progression [3]. However, notwithstanding these encouraging results, we are still far from benefiting from an effective treatment, and we urgently need a better understanding of the pathophysiological processes of AD—potentially related to Aβ peptides—in order to develop complementary and original therapeutics approaches. 

Alterations in neuronal electrical activity and network oscillations are among the first signals in the brains of AD patients, and they are intimately associated with Aβ deposits even prior to the onset of clinical symptoms [4,5,6,7,8]. In rodents, treatment with soluble Aβ peptides can lead to neuronal hyper-excitation [9,10], and modulation of excitatory synaptic activity in turn regulates the proteolysis of APP and release of Aβ peptides [11,12], creating positive feedback between Aβ accumulation and neuronal electrical activity/synapse dysfunction. Early impairment of neuronal circuits is also observed in the brains of transgenic AD animal models, including the APP23xPS45 double-transgenic mice [13] and the 5XFAD mice, which overexpress the human amyloid precursor protein (APP) and presenilin 1 (PS1) harboring five familial AD mutations [14,15]. Moreover, human-induced pluripotent stem cells (hiPSC)-derived neurons carrying mutations in *APP* or *PSEN1* show altered electrical properties [16], suggesting that Aβ peptides modulate the functional properties of human neurons. However, the precise molecular mechanisms associated with Aβ-mediated neuronal hyper-excitability and the exact contribution of Aβ_1–42_ peptides to this process remain elusive. 

In this work, we investigate the early functional and gene expression alterations in human-induced neurons (hiNs) and astrocytes (hiAs) exposed to low concentrations of cell-secreted Aβ peptides. We show that a single exposure to low concentrations of Aβ peptides is sufficient to promote an increase in the frequency of calcium transients in hiNs associated with specific synapse-related gene expression alterations in glutamatergic neurons. These results suggest that slight changes in Aβ concentrations in the brain may have long-lasting consequences for neural networks by impacting gene expression mainly in glutamatergic neurons. 

## 2. Materials and Methods

### 2.1. Maintenance of hiPSCs and Neural Induction

For this project, we took advantage of a commercially available hiPSC cell line (ASE-9109, Applied StemCell Inc., Milpitas, CA, USA). The maintenance of the iPSC cultures was performed in adherence with the manufacturer’s protocols, which can be found on the webpage of Stemcell Technologies. Briefly, hiPSCs were maintained in mTeSR1+5X supplemented medium in non-treated cell culture dishes/plates pre-coated with 10 ug/mL of vitronectin freshly diluted in Cell Adhere Dilution Buffer (StemCell Technologies Inc., Vancouver, BC, Canada) for 1 h at room temperature. Full-medium change was performed daily until the confluence reached about 80%, when the cells were passaged. Cell number and viability were recorded using a LUNA™ Automated Cell Counter. To induce a neural lineage from the iPSCs, we used the embryoid body protocol validated by StemCell Technologies and based on the dual SMAD inhibition (SMADi) method [17]. Briefly, hiPSCs are seeded on AggreWell plates (StemCell Technologies) to produce the embryoid bodies, which are then replated in classic plates followed by a Neural Rosette selection to retain the neural cells. After this period, the hiNPCS generated are maintained for up to 9 passages in treated cell culture dishes pre-coated with poly-L-ornithine (PLO) and laminin (10 μg/mL), using supplemented STEMdiff™ Neural Progenitor Medium (NPM; StemCell Technologies Inc., Vancouver, BC, Canada), which is fully changed daily. The PLO solution was produced in water (0.001%), while laminin was diluted in PBS 0.01M with Ca^2+^ and Mg^2+^.

### 2.2. Differentiation of hiNPCs and Mixed Cultures of hiNs and hiAs

To obtain mixed 2D cultures comprising hiNs and hiAs from the hiNPCs, we followed the spontaneous differentiation protocol detailed by StemCell Technologies. Following this spontaneous protocol, 100,000 hiNPCs/well were plated in 24-well cell imaging plates from Eppendorf (Cat # 0030741005) pre-coated with PLO (0.001%) and laminin (10 µg/mL). Cells were kept in 0.5 mL of NPC medium per well for 24 h. Following this, an equal volume of complete BrainPhys medium supplemented with BDNF, GDNF, laminin, dibutyryl-cAMP, ascorbic acid, N2 and SM1 (StemCell Technologies Inc., Vancouver, BC, Canada) was added to each well to begin the process of differentiation [18]. Subsequently, the media were changed in the plates bi-weekly. The media change consisted of removing half of the existing medium from each well and replacing it with an equal volume of fresh complete BrainPhys medium. Mixed cultures of hiNs and hiAs were exposed to cell-secreted or synthetic Aβ peptides (see below) after 5–6 weeks of differentiation and then processed for calcium imaging, Western blotting, immunocytochemistry, or snRNA sequencing.

### 2.3. Cell-Secreted Aβ Peptides Treatment

Chinese hamster ovary (CHO) cell lines overexpressing the human APP695 (CHO-APP^WT^) or the London mutated APP695 (CHO-APP^V717L^) were previously described [19]. CHO cells were grown in DMEM/F-12 1:1 medium, supplemented with 10% heat-inactivated fetal bovine serum, 0.2% Pen/Strep, 2% HT supplement and 300 µM Proline (Sigma, Kawasaki, Japan). One day before CHO supernatant media collection, a full-medium replacement was performed with BrainPhys. CHO cell conditioned BrainPhys medium was collected 24 h later, and Aβ concentrations were measured using Alpha-LISA kits specific for human Aβ_1–X_ (AL288C, PerkinElmer, Waltham, MA, USA) and Aβ_1–42_ (AL276C, PerkinElmer). Briefly, the human Aβ analyte standard was diluted in the BrainPhys medium. For the assay, 2 µL of cell culture medium or standard solution was added to an Optiplate-384 microplate (PerkinElmer). A volume of 2 µL of the 10X mixture, including acceptor beads and the biotinylated antibody, was then added to the wells with culture media or standard solution. Following incubation at room temperature for 1 h, 16 µL of 1.25X donor beads was added to the respective wells and incubated at room temperature for 1 h. Luminescence was measured using an EnVision-Alpha Reader (PerkinElmer) at 680 nm excitation and 615 nm emission wavelengths. Control treatment was performed using unconditioned/blank BrainPhys medium.

### 2.4. Synthetic Aβ_1–42_ Treatment

Commercially available lyophilized Aβ_1–42_ (Anaspec, Fremont, CA, USA) was reconstituted in 1% DMSO as the solvent, followed by PBS 10 mM at a concentration of 1 µM and re-diluted in 500 µL of supplemented BrainPhys medium to reach a final concentration of 100 pM. The inverted control peptide (Aβ_42–1_, Anaspec) at the same final concentration and vehicle only (1% DMSO in PBS 10 mM) were used as controls. After 5 weeks of spontaneous differentiation, half of the medium was removed from each well and kept aside for later replacement. Then, either the Aβ_1–42_, the inverted peptide or the vehicle were added at proper concentrations, and the cells were kept in the incubator for 24 h. After that period, the medium in each well was replaced, followed by the calcium imaging procedures and protein extraction.

### 2.5. Calcium Imaging Experiments

Spontaneously differentiated 5–6-week cultures containing hiNs and hiAs were treated with cell conditioned medium or synthetic Aβ_1–42_ (and respective controls) for 48 h and 24 h, respectively, prior to real-time calcium imaging recordings. Cells were incubated with Oregon Green™ 488 BAPTA-1 (OGB-1) acetoxymethyl (AM) (ThermoFisher Scientific, Waltham, MA, USA) for 1 h immediately before the recordings. A 2.5 mM stock solution of the calcium-indicator dye was prepared in Pluronic™ F-127 (20% solution in DMSO) (ThermoFisher Scientific). A volume of 1 μL of the dye solution was added to 500 μL of existing BrainPhys medium in each well of a 24-well cell imaging plate. The other half of existing BrainPhys media was removed from the wells of the plate and kept aside at 37 °C, while the calcium-indicator dye was incubated in fresh BrainPhys medium. After 1 h of incubation, the medium, which was kept aside, was replaced in each well. The 2D cultures were then ready to be imaged using a Spinning Disk Microscope (Nikon) housed at the Institut Pasteur de Lille, Lille, France, using the MetaMorph imaging software. We took 1000 images using a 20× (synthetic peptide treatment) or 40× (conditioned medium treatment) objective with 10 ms exposure time and 200 ms intervals, totaling up to 3 min of recording per field. For each well, up to 5 random fields were chosen, and at least two wells from 3 (for cell conditioned medium) or 4 (for synthetic Aβ) independent culture batches were imaged for each condition.

### 2.6. Analyses of Calcium Transients

The time-series data from calcium imaging recordings were at first converted to the .avi format after background subtraction using ImageJ v1.53 (National Institute of Mental Health, Bethesda, MD, USA). The videos were subsequently opened in the CALciumIMagingAnalyzer (CALIMA) software [20]. Each video recording of a field of cells was first downscaled to 2× in terms of size with a 10× zoom and was checked for the frame average mode. Moreover, in this first detection stage, pre-set filter parameters were adjusted and applied to enable the detection of the maximum number of fluorescent neurons (regions of interest—ROIs) in each field. During this step, the detected cells were checked several times to ensure that only neurons were selected, based both on cell morphologies and the presence of fast calcium transients. Fluorescence signal noise was filtered using a median of 3 consecutive images each. Fluorescence changes in all detected ROIs were then recorded. Fast (<400 ms) increases in the fluorescence signal above 2 or more standard deviations of the average background were considered a calcium transient (or spike). The detected spikes and cross-correlation matrices of all cells from each field of imaging were extracted and exported as .csv files. To measure the synchronicity of calcium transients among the hiNs, we calculated the Eigenvalues of cross-correlation matrices generated by crossing the activity of each individual neuron with all other neurons in the same image field [21].

### 2.7. Single-Nucleus RNA Sequencing

Six-week-old mixed cultures of hiNs and hiAs were washed in 24-well plate wells with 1 mL of Deionized Phosphate Buffer Saline 1X (DPBS, GIBCO™, Fisher Scientific 11590476). Cells were lysed with wide bore tips in 1 mL Lysis Buffer (Tris-HCL 10 mM, NaCl 10 mM, MgCl2 3 mM, Tween-20 0.1%, Nonidet P40 Substitute 0.1%, Digitonin 0.01%, Invitrogen™ RNAseout™ recombinant ribonuclease inhibitor 0.04 U/μL). Multiple mechanical resuspensions in this buffer were performed for a total lysis time of 15 min. A volume of 500 μL of washing buffer was added (Tris-HCL 10 mM, NaCl 10 mM, MgCl_2_ 3 mM, Tween-20 0.1%, BSA 1%, Invitrogen™ RNAseout™ recombinant ribonuclease inhibitor 0.04 U/μL), and the lysis suspension was centrifuged for 8 min at 500× *g* at 4 °C (the same protocol was used for all the following centrifugation steps). Nuclei pellets were washed two times using the washing buffer with one filtration step involving MACS pre-separation filter 20 μm (Miltenyi Biotec, Bergisch Gladbach, Germany). Nuclei pellets were resuspended in 100 μL of staining buffer (DPBS BSA 2%, Tween-20 0.01%, 0.04 U/μL Invitrogen™ RNAseout™ recombinant ribonuclease inhibitor), 10 μL of Fc blocking reagent HumanTruStainFc™ (422302, Biolegend, San Diego, CA, USA) and incubated for 5 min at 4 °C. A volume of 1 μL of anti-Vertebrate Nuclear Hashtag Antibody (Total-Seq™-A, Biolegend) was added and incubated for 15 min at 4 °C. Nuclei pellets were washed three times in the staining buffer with one filtration step involving MACS pre-separation filter 20 μm (Miltenyi Biotec) to a final resuspension in 300 μL of staining buffer for Malassez cell counting with Trypan blue counterstaining (Trypan Blue solution, 11538886, Fisher Scientific). The isolated nuclei were loaded on a Chromium 10X Genomics controller, following the manufacturer’s protocol, using the chromium single-cell v3 chemistry and single indexing and the adapted protocol by Biolegend for the HTO library preparation. The resulting libraries were pooled at equimolar proportions at a 9 to 1 ratio for the gene expression library and the HTO library, respectively. Finally, the pool was sequenced using 100 pb paired-end reads on the NOVAseq 6000 system, following the manufacturer’s recommendations (Illumina, San Diego, CA, USA). 

The unique molecular index (UMI) count matrices for the gene expression and hashtag oligonucleotide (HTO) libraries were generated using the CellRanger count (feature barcode) pipeline. The reads were aligned on the GRCh38-3.0.0 transcriptome reference (10x Genomics Pleasanton, CA, USA). Filtering for low-quality cells based on the number of RNA, the genes detected and the percentage of mitochondrial RNA was performed. For the HTO sample, the HTO matrix was normalized using centered log-ratio (CLR) transformation, and cells were assigned back to their sample of origin using the HTODemux function of the Seurat R Package. Then, normalizations of the gene expression matrix for cellular sequencing depth, mitochondrial percentage and cell cycle phases, using the variance stabilizing transformation (VST)-based Seurat::SCTransform function, were performed.

To integrate the experimental replicates of single-cell experiments, the harmony R package [22] was used. In order to integrate the datasets, the SCTransform normalized matrices were merged, and PCA was performed using the Seurat::RunPCA default parameter. The 50 principal components (dimensions) of the PCA were corrected for batch effect using the harmony::RunHarmony function. Then, the first 30 batch-corrected dimensions were used as input for the graph-based cell clustering and visualization tool. The Seurat::FindNeighbors function, using the default parameters, and the Seurat::FindClusters function, using the Louvain algorithm, were used to cluster cells according to their batch-corrected transcriptome similarities. To visualize the cell similarities in a two-dimensional space, the Seurat::RunUMAP function, using the default parameter, was used. Cell clusters were then annotated based on cell-type-specific gene expression markers.

To study the effect of conditioned media from CHO cells expressing APP WT or APP V717L on hiNPCs-derived cells, cell-type-specific differential expression analysis was performed on 7578 cells (2570 “APPV717L” cells, 2262 “APPWT” cells and 2746 control cells) using the Wilcoxon rank-sum test after normalization and regularized variance stabilization of the raw count using SCTransform [23]. We used an adjusted *p*-value < 0.05 and |log_2_FC| > 0.25 as cut-offs to define the differently expressed genes. Functional enrichment analysis was performed using the R package fast pre-ranked gene set enrichment analysis (FGSEA, version 1.26.0). To perform the FGSEA, we utilized a gene set list with an adjusted *p*-value < 0.05 sorted based on the log2FC from three comparisons: APPV717L x Control, APPV717L x APPWT and APPWT x Control. For the FGSEA multi-level function, the minimum gene set size was set to 15, while the maximum size was limited to 400. The significance threshold for *p*-values was defined as *p*-value < 0.05. To determine the up- or down-regulated terms within the considered ontologies among the conditions, we employed the normalized enrichment score (NES). Terms with NES values greater than 0 were classified as up-regulated, while those with NES values less than 0 were classified as down-regulated. The FGSEA multi-level analysis was performed across all cell types identified within the conditions.

### 2.8. Statistical Analysis

All statistical analyses and preparation of graphs were performed using Graph-Pad Prism 8 (San Diego, CA, USA) or RStudio (RStudio Team (2020)) (RStudio: Integrated Development for R. RStudio, PBC, Boston, MA, USA (http://www.rstudio.com/)). For all statistical comparisons, we first tested for the normality of data distribution using the Kolmogorov–Smirnov test and then compared group variances using the appropriate parametric or non-parametric tests, as indicated in figure legends. Data in the plots are represented as mean + standard deviation (SD). An α-error level of *p* < 0.05 was considered significant.

## 3. Results

### 3.1. Exposure to Low Concentrations of Aβ Peptides Increases Electrical Activity and Reduces Synchronicity of hiNs

Increased concentrations of Aβ can lead to neuronal hyper-excitation [9,10,13] and reduced synaptic connectivity in AD animal models. To evaluate whether increased concentrations of Aβ could also affect human neurons, we treated 6-week-old mixed hiNs/hiAs cultures with cell-secreted Aβ peptides (Figure 1A). First, we measured the concentrations of soluble Aβ peptides in the conditioned medium derived from CHO cells expressing the APP^V717L^ (CHO-APP^V717L^) or APP^WT^ (CHO-APP^WT^) using alphaLISA and observed average concentrations of 700 pM Aβ_1–x_ and 350 pM Aβ_1–42_ in CHO-APP^WT^ conditioned medium (Figure 1B). As previously described [19,24], the concentration of Aβ_1–x_ was 8-fold higher, whereas that of Aβ_1–42_ was 2.5-fold higher in CHO-APP^V717L^ compared to CHO-APP^WT^ conditioned medium (Figure 1B).

Forty-eight hours after treatment with conditioned medium, we labeled cells with the calcium-sensitive dye Oregon green BAPTA and performed time-lapse video microscopy (Figure 2A,B). We observed a significantly higher frequency of calcium spikes in hiNs cultures treated with conditioned medium derived from CHO-APP^V717L^ compared to that from CHO-APP^WT^ (Figure 2C). To evaluate whether this change in the frequency of calcium spikes could be explained by the higher concentration of Aβ_1–42_ in the conditioned medium from CHO-APP^V717L^ (Figure 1B), we performed similar calcium imaging experiments in hiNs/HiAs cultures 24 h after treatment with 100 pM of synthetic Aβ_1–42_ or its inverted control peptide (ICP). We observed a significantly higher frequency of calcium spikes in hiNs treated with Aβ_1–42_ compared to both vehicle and ICP (Figure 2D). The proportion of active cells (defined as cells showing at least one calcium spike detected during the time of observation) was similar among all conditions (cell-secreted Aβ—Control: 44.36 ± 22.27%; APP WT: 39.68 ± 15.5%; APP V717L: 42.21 ± 20.71%; synthetic Aβ—Control: 31.31 ± 9.59%; ICP: 39.61 ± 6.85%; Aβ_1–42_: 36.2 ± 11.78%). However, we observed a lower cross-correlation among inter-neuron calcium spikes in cultures exposed to cell-secreted Aβ or synthetic Aβ_1–42_ (Figure 2F,G), suggesting a reduction in neuronal synchronization. Altogether, these observations indicate that exposure to low concentrations of cell-secreted Aβ or synthetic Aβ_1–42_ peptides is sufficient to alter the electrical activity of hiNs circuits. 

### 3.2. Treatment with Conditioned Media Containing Higher Concentrations of Aβ Peptides Elicits Transcriptional Changes Mainly in hiNs

We next sought to identify the possible changes in gene expression in human-induced astrocytes and neurons elicited by treatment with Aβ. To that end, we performed snRNA sequencing of mixed hiNs/hiAs cultures treated with conditioned media from CHO-APP^V717L^ or CHO-APP^WT^ and blank BrainPhys (controls) for 48 h (Figure 1). After quality control (Section 2), we recovered 7578 nuclei and identified 17 different cell types/subtypes using unsupervised clustering based on gene expression (Figure 3A). We observed that treatment with conditioned media significantly affected the proportion of cells in clusters 1 and 3, but there was no difference between CHO-APP^V717L^ and CHO-APP^WT^ (Figure 3B). To annotate the main cell types in our dataset, we identified the top markers of each cluster (Appendix A) and evaluated the expression of the astrocyte markers SLC1A3 (GLAST), GFAP and TNC; pan-neuronal markers SNAP25, DCX, MAPT; glutamatergic neuron marker SLC17A6 (VGLUT2); GABAergic neuron markers GAD1 and GAD2; and neural progenitor cell (NPC) markers HES6, CCND2 and CDK6 in different clusters (Figure 3C). We were able to identify two NPCs (14 and 17), four astrocytes (2, 3, 4 and 5), seven glutamatergic neurons (12, 6, 7, 9, 10, 11 and 12) and three GABAergic neurons clusters (8, 12 and 15). We were unable to unambiguously assign cluster 16 to any specific cell type based on its gene expression profile and therefore annotated it as “unclassified” (Figure 3D). Although we observed significant statistical differences in the proportions of cell types within the same condition, no differences were observed when we compared the proportion of an individual cell type among conditions (Figure 3E). We also evaluated the expression of putative Aβ ligands [25] in our cultures and found that hiNs mainly expressed acetylcholine receptor subunit alpha-7 (CHRNA7), glutamate ionotropic receptor NMDA type subunit 1 (GRIN1), glutamate metabotropic receptor 5 (GRM5) and LDL receptor related protein 1 (LRP1), whereas hiAs mainly expressed LRP1 and prion protein (PRNP) (Figure 3F). 

To characterize early molecular alterations elicited by conditioned media treatments, we used the Wilcoxon rank-sum test to identify the differently expressed genes (DEGs) in NPCs, astrocytes, glutamatergic and GABAergic neurons independently. We found that treatment with conditioned medium from CHO-APP^WT^ significantly (log2FC > 0.25 and FDR < 0.05) affected the expression of 252 genes in astrocytes, 32 genes in GABAergic neurons and 89 genes in glutamatergic neurons (Figure 4A; Appendix A). Nevertheless, a more significant effect was observed in cultures treated with conditioned medium from CHO-APP^V717L^, where we identified 235 DEGs in astrocytes, 228 in GABAergic neurons and 596 in glutamatergic neurons (Figure 4B; Appendix A). Interestingly, a direct comparison of the transcriptional profile of cultures treated with conditioned medium from CHO-APP^V717L^ or CHO-APP^WT^ revealed a marked alteration in gene expression of glutamatergic neurons (174 DEGs in APP V717L compared to APP WT) but only 2 DEGs in GABAergic neurons and 10 DEGs in astrocytes (Figure 4C; Appendix A). Together, these data indicate that elevated levels of APP and its metabolites (present in both conditions using conditioned media as compared to controls) modulate gene expression in astrocytes and different neuronal subtypes and that further increase in Aβ concentrations (observed in CHO-APP^V717L^ compared to CHO-APP^WT^ conditioned medium) has a more prominent effect on gene expression of glutamatergic neurons.

### 3.3. Genes Modulated by Aβ Peptides Are Mainly Associated with Oxidative Stress and Synapse Transmission

Next, we performed a gene set enrichment analysis (GSEA; Subramanian 2005) to identify the functional and pathway enrichments modulated by conditioned media treatments in each individual cell type. We observed significant functional enrichments for the gene sets identified in the comparison of CHO-APP^V717L^ vs. control but not CHO-APP^WT^ vs. control conditions (Figure 5A,B). In astrocytes, treatment with the conditioned medium from CHO-APP^V717L^ up-regulated genes associated with the regulation of cell differentiation and down-regulated genes associated with the catabolic process and endoplasmic reticulum lumen (Appendix A). In neurons, up-regulated genes were mainly associated with synapse-related terms, whereas down-regulated genes were associated with metabolic processes (Figure 5A,B; Appendix A). We also found functional enrichment for gene sets associated with neuronal intrinsic electrical properties, such as ion channels and calcium ion binding, in glutamatergic but not GABAergic neurons treated with CHO-APP^V717L^ conditioned medium compared to controls (Figure 5A; Appendix A). Notably, only glutamatergic neurons displayed functional enrichments for the gene sets identified in cultures treated with conditioned medium from CHO-APP^V717L^ compared to CHO-APP^WT^ (Figure 5C; Appendix A), further suggesting a more prominent effect of Aβ on gene expression of this cell type. Up-regulated genes were enriched in terms of cell adhesion, regulation of cellular component biogenesis and post-synapse membrane, whereas down-regulated genes were enriched in terms of transport activity, mitochondrion and catalytic complex (Figure 5C; Appendix A). 

### 3.4. Selective Effect of Aβ on Synaptic Processes of Glutamatergic Neurons

Synapses are the fundamental information processing units of the brain, and synaptic dysregulation is central to AD pathology. To further characterize our transcriptomic data in a synapse context, we used SynGO, a systematic annotation tool for synaptic genes and the ontology of synaptic processes [26]. Confirming our previous observations using GSEA, we observed the enrichment of several gene ontologies (GOs) related to synaptic biological processes in both glutamatergic and GABAergic neurons (Figure 6; Appendix A). This enrichment was mainly evident for DEGs identified in neurons exposed to conditioned medium from CHO-APP^V717L^ compared to controls (Figure 6A,B)—affecting both the pre- and post-synaptic compartments—and associated with the regulation of pre-synaptic cytosolic calcium levels, synaptic vesicle cycle, regulation of post-synaptic membrane neurotransmitter receptor levels and synapse assembly, among other synaptic processes (Appendix A). Additionally, as in our GSEA analysis, we only observed significant enrichments for DEGs identified in glutamatergic neurons when comparing conditioned medium CHO-APP^V717L^ vs. CHO-APP^WT^ (Figure 6C; Appendix A).

### 3.5. Exposure to Aβ Peptides Alters Neuroglial Communication

Lastly, we leveraged our snRNA sequencing data to infer the inter-cellular communication patterns in our hiNs/hiAs cultures and to evaluate how these patterns could be impacted by the treatment with CHO-APP^V717L^ or CHO-APP^WT^ conditioned medium. To that end, we employed CellChat to predict the major cell-to-cell signaling pathways conserved and context-specific across different conditions [27]. We observed consistent signaling patterns in astrocytes, glutamatergic neurons and GABAergic neurons in all three experimental conditions, but some conspicuous differences could also be observed (Figure 7; Appendix A). Compared to controls, both CHO-APP^V717L^ and CHO-APP^WT^ conditioned media inhibited neuregulin (NRG), epidermal growth factor (EGF) and myelin protein zero (MPZ) signaling in astrocytes, stimulated cadmium (CADM) signaling in glutamatergic neurons and CD46 signaling in astrocytes, glutamatergic and GABAergic neurons (Figure 7A,B). We also observed specific changes in the signaling patterns of cells treated with CHO-APP^V717L^ compared to CHO-APP^WT^ conditioned media, including a higher relative strength of ephrin-A (EPHA), semaphorin (SEMA5) and NECTIN signaling, and a lower relative strength of midkine (MK; also known as neurite-growth-promoting factor 2—NEGF2) and fibronectin (FN1) signaling (Figure 7C). Consistent with the predominant up-regulation in expression observed in hiNs (Figure 4), we also found that treatment with CHO-APP^V717L^ or CHO-APP^WT^ conditioned media increased both the outgoing and incoming interaction strength in the aggregated cell–cell communication network from all signaling pathways in glutamatergic neurons and, to a lesser extent, in GABAergic neurons (Figure 7D). Similarly, we observed a reduced outgoing and incoming interaction strength in astrocytes (Figure 7D), which also showed a more prominent down-regulation of genes after conditioned medium treatments (Figure 4). Collectively, these analyses indicate that small increases in Aβ concentrations may have important consequences for neuroglial communication.

## 4. Discussion

AD patients show aberrant neural network activity in the form of both hyper- and hypo-excitability, and several lines of evidence indicate that these network alterations result from changes in neuronal excitability and synaptic transmission tied to elevated Aβ levels [13,28,29,30]. Yet, the molecular mechanisms underlying Aβ-induced hyper-excitability remain unclear. In this work, we show that human-induced neurons acutely exposed to low concentrations of both cell-secreted Aβ peptides (including Aβ_1–42_) or synthetic Aβ_1–42_ present alterations in the frequency of calcium transients suggestive of increased neuronal excitability. We also show that cell-secreted Aβ up-regulates the expression of several synapse-related genes and down-regulates the expression of genes associated with metabolic processes in mitochondria mainly in glutamatergic neurons and to a lesser degree in GABAergic neurons and astrocytes. Collectively, these findings suggest that slight elevations in Aβ concentrations are sufficient to elicit transcriptional changes in human neurons, which can contribute to early alterations in neural network activity. 

Aβ peptides play a central role in the pathogenesis of AD. When they accumulate abnormally in the brain, they can have detrimental effects on neurons, such as the induction of neuritic plaques, disruption of calcium homeostasis, synaptic loss and oxidative stress [31,32], thus directly contributing to neuronal dysfunction and death. Aβ peptides also affect astrocytes [14,33,34,35], which can indirectly contribute to neuronal dysfunctions in AD. Moreover, the presence of Aβ in the brain can activate microglia—the immune cells of the central nervous system—leading to neuroinflammation [36,37]. Chronic neuroinflammation can further exacerbate neuronal damage and contribute to the progression of AD. Therefore, Aβ-mediated signaling mediates both cell-autonomous and non-cell-autonomous processes impacting neuronal function and neurodegeneration. In our work, we only model the possible effects of Aβ peptides in neurons and astrocytes. Future work should evaluate the effects of those peptides in more complex hiPSC-derived cellular systems harboring neurons, astrocytes, oligodendrocytes, microglia and endothelial cells, thus better representing the complex cellular landscape of the brain.

Despite this limitation, our results reveal some important features of the acute neuronal response to low concentrations of APP and Aβ peptides, which can partly mirror the early stages of AD pathology. Here, we show that exposure to CHO-APP^V717L^ conditioned medium—which mainly differs from CHO-APP^WT^ conditioned medium in the higher concentration of Aβ oligomers enriched for Aβ_1–42_ [19,24]—leads to a higher frequency and lower synchronicity of calcium spikes in hiNs. We also describe similar alterations in cultures treated with purified Aβ_1–42_ peptide, suggesting that acute exposure to a low concentration of Aβ_1–42_ is sufficient and necessary to induce human neuron hyper-excitability and reduced coordination of network activity. These findings agree with previous studies showing that hiPSC-derived neurons bearing AD-related mutations in presenilin-1 or amyloid precursor protein show a higher frequency of spontaneous action potential compared to isogenic gene corrected controls, which is reversed via either γ-Secretase or BACE1 inhibition [16]. The findings also coincide with studies in mice showing that low concentrations (200 pM) of Aβ_1–42_ induce an increase in the frequency of miniature EPSCs, a decrease in paired pulse facilitation, an increased length of post-synaptic density and an increased expression of plasticity-related proteins [9,28]. These effects were present upon extra-cellular but not intra-cellular application of the peptide and involved the activation of α7 nicotinic acetylcholine receptors [28], which were also expressed in a fraction of hiNs (mainly glutamatergic neurons) in our cultures. 

The alterations in neuronal calcium dynamics, together with the gene expression pattern of known Aβ_1–42_ ligands observed in glutamatergic and GABAergic neurons in our cultures, may suggest a direct effect of this peptide on the regulation of neuronal functional properties. However, our experimental design does not allow us to unambiguously disentangle the direct effects of Aβ peptides in neurons from potential indirect effects mediated by astrocytes. In fact, the inclusions of aggregated Aβ in astrocytes critically affect their ability to support neuronal function [35], and the recovery of astrocytic calcium activity normalized neuronal hyper-activity in a mouse model of amyloidopathy [34], suggesting that at least some of the neuronal alterations observed in response to Aβ peptides are mediated by astrocytes. Yet, our transcriptomic data indicate that exposure to Aβ peptides directly modulates gene expression in glutamatergic neurons, since only this cell type showed significant alterations in gene expression when exposed to CHO-APP^V717L^ compared to CHO-APP^WT^ conditioned medium. Moreover, the higher expression of putative Aβ receptors in hiNs compared to hiAs, together with the predominant number of genes differentially expressed in glutamatergic neurons exposed to CHO-APP^V717L^ conditioned medium compared to GABAergic neurons or astrocytes, suggest a primarily cell-autonomous effect of Aβ-mediated signaling in glutamatergic neurons. 

Our transcriptional data also reveal critical effects of Aβ-mediated signaling on the regulation of metabolism, electrical activity and synaptic plasticity in hiNs, especially glutamatergic ones. These finding are in accordance with previous studies in AD animal models at the early stages of amyloidopathy [38,39,40] and in the brains of patients with AD at the early Braak stages [38,41,42,43,44]. Thus, we believe that our results in hiNs may provide insightful information on the molecular mechanisms mediating Aβ-mediated neuronal hyper-excitability in AD. Moreover, our innovative approach in identifying inter-cellular signaling pathways impacted by Aβ peptides via the identification of interactions among ligands, receptors and their co-factors expressed by hiNs and hiAs may contribute to unraveling new targetable molecules to prevent Aβ-mediated toxicity. 

We show that treatment with CHO-APP^V717L^ conditioned medium activates EPHA, SEMA5 and NECTIN signaling mainly in glutamatergic neurons and, to a lesser extent, in GABAergic neurons and astrocytes. Ephrin ligands and their associated Eph receptors guide axons during neural development and regulate synapse formation and neuronal plasticity in the adult [45]. Similarly, members of the nectin and semaphorin signaling pathways regulate several aspects of synapse formation and maintenance during development and adulthood [46,47]. Among the pathways activated in hiNs treated with CHO-APP^V717L^ compared to CHO-APP^WT^ conditioned medium, SEMA5B regulates the elimination of synaptic connections in cultured hippocampal neurons [48], while EPHA1 is a risk-modifying locus for AD [49,50]. Therefore, it is plausible to speculate that at least some of the functional and gene expression alterations observed in hiNs exposed to Aβ peptides are mediated by those signaling pathways. 

Although we cannot categorically rule out the contribution of soluble APP products—rather than Aβ peptides—to the alterations observed in the activity and gene expression of hiNs, the fact that they are mainly observed in cells treated with CHO-APP^V717L^ conditioned medium strongly suggests that soluble Aβ peptides are the key regulators of neuronal plasticity. On the other hand, the milder alterations observed in the expression of synapse-related genes of hiNs exposed to the CHO-APP^WT^ conditioned medium could suggest that even lower amounts of Aβ peptides, APP itself or other APP metabolites also affect these cells, as previously suggested in AD mouse models [9,51]. Future studies could employ our system to evaluate the impact of Aβ and other APP metabolites at different concentrations and aggregation states on the regulation of the functional properties and gene expression in human neurons, thus further advancing our understanding of the physiological and pathological roles of these complexes signaling molecules.

## Figures and Tables

**Figure 1 biomedicines-11-02564-f001:**
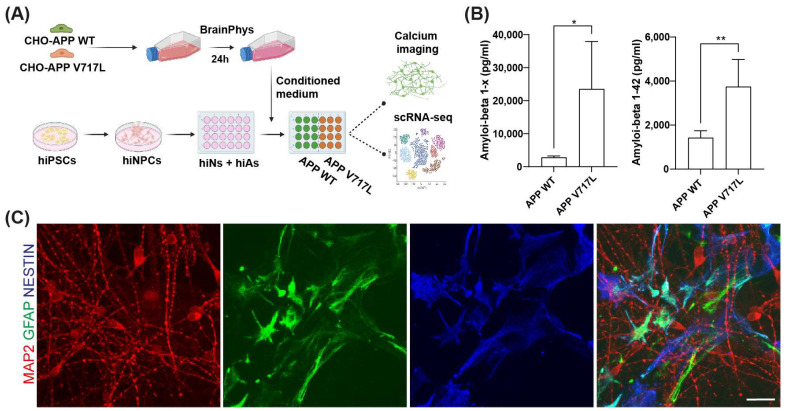
Different concentrations of Aβ peptides in CHO-APPWT and CHO-APPV717L conditioned media. (**A**) Experimental design showing the generation of hiNs/hiAs cultures, medium conditioning and treatment. (**B**) Quantification of Aβ_1–x_ and Aβ_1–42_ peptides in the conditioned medium from CHO-APPWT or CHO-APPV717L (* *p* < 0.05; ** *p* < 0.01; Unpaired *t*-test). (**C**) Immunocytochemistry in 6-week-old hiNs/hiAs culture using antibodies against MAP2 (red), GFAP (green) and NESTIN (blue). Calibration bar: 20 μm.

**Figure 2 biomedicines-11-02564-f002:**
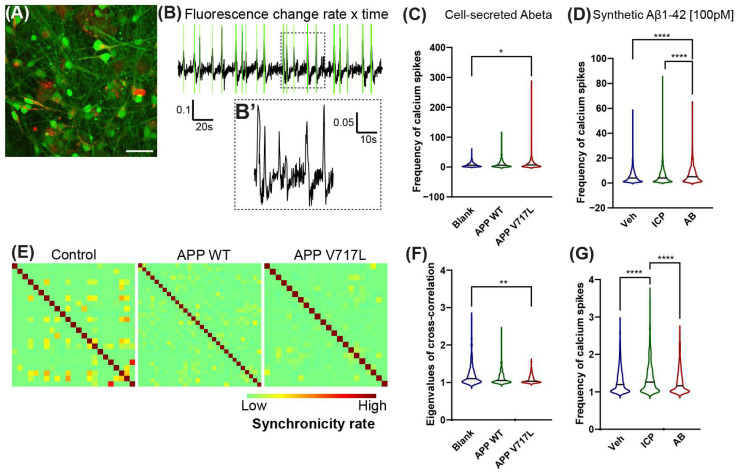
Altered frequency of calcium transients in hiNs exposed to Aβ peptides. (**A**) Snapshot of 6-week-old hiNs/hiAs culture labeled with Oregon green BAPTA (green) and treated 24 h before with 100 nM of HiLyte™ Fluor 555-labeled Aβ_1–42_ (red). Calibration bar: 50 μm. (**B**,**B’**) Representative plot of fluorescence change over time in 1000 frames. Green vertical lines indicate detected calcium spikes (∆F > 2 SDs). Dashed box is magnified in (**B’**). (**C**) Quantification of calcium spikes in cultures treated with conditioned medium 48 h before imaging. (**D**) Quantification of calcium spikes in cultures treated with 100 pM synthetic Aβ_1–42_ 24 h before imaging. (**E**) Representative cross-correlation matrices showing the synchronicity rate of hiNs calcium transients in different conditions. (**F**) Graphic showing the Eigenvalues of cross-correlation matrices in control and cell-secreted treated hiNs/hiAs cultures. (**G**) Same for cultures treated with synthetic Aβ_1–42_. Statistics in (**C**,**D**), (**F**,**G**): * Padj < 0.05; ** Padj < 0.01; **** Padj < 0.0001; Kruskal–Wallis followed by Dunn’s multiple comparison test; *n* = 3 independent cultures for each condition. Number of analyzed cells per condition: Blank Brainphys = 1098; APP^WT^ = 580; APP^V717L^ = 670; Vehicle = 2543; ICP = 3459; Aβ_1–42_ = 2472.

**Figure 3 biomedicines-11-02564-f003:**
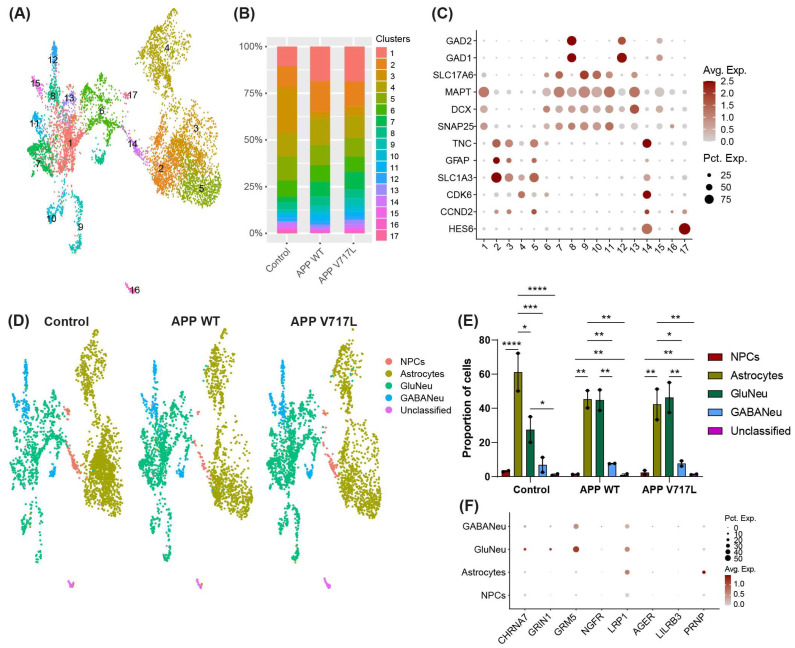
Similar cellular composition of hiNs/hiAs cultures treated with control or CHO cells conditioned media. (**A**) UMAP representation of the different cell clusters in hiNs/hiAs cultures. (**B**) Relative proportion of cell clusters. (**C**) Dot plot showing the expression of key cell markers used to annotate cell populations. (**D**) UMAP representation of the annotated cell types/subtypes in hiNs/hiAs cultures in different treatment conditions. (**E**) Proportion of cell types/subtypes in different conditions (statistics). * Padj < 0.05; ** Padj < 0.01; *** Padj < 0.001; **** Padj < 0.0001. (**F**) Dot plot showing the expression of putative Aβ ligands in different cell types.

**Figure 4 biomedicines-11-02564-f004:**
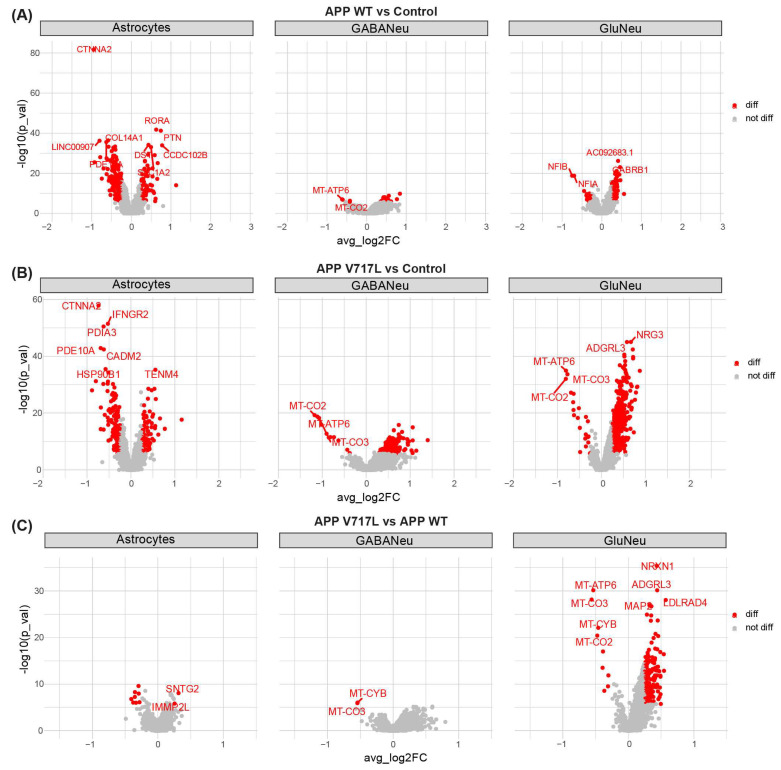
Exposure to Aβ peptides modifies gene expression in hiNs and hiAs. (**A**) Volcano plots representing DEGs (red dots, adjusted *p*-value < 0.05 and |log2FC| > 0.25) identified in astrocytes, glutamatergic and GABAergic neurons when comparing CHO-APP^WT^ vs. control or CHO-APP^V717L^ vs. control and CHO-APP^V717L^ vs. CHO-APP^WT^.

**Figure 5 biomedicines-11-02564-f005:**
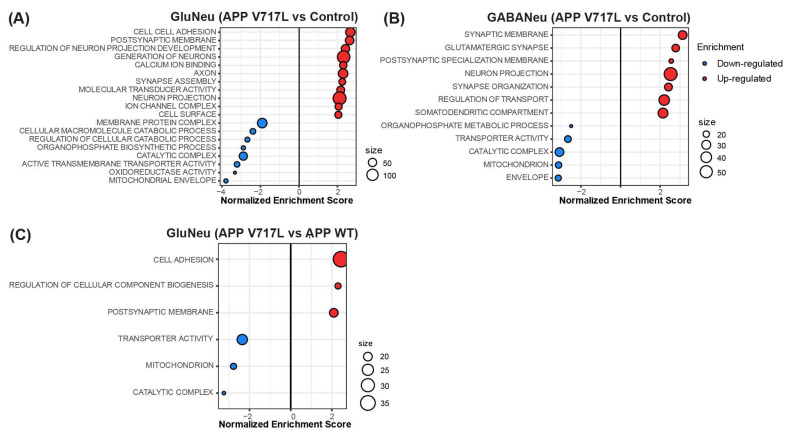
Enrichment of synaptic and metabolic pathways in hiNs treated with Aβ peptides. (**A**–**C**) Dot plots showing the normalized enrichment scores (NES) for pathways enriched in genes differentially expressed in glutamatergic neurons (**A**,**C**) and GABAergic neurons (**B**) when comparing CHO-APP^V717L^ vs. control (**A**,**B**) or CHO-APP^V717L^ vs. CHO-APP^WT^ (**C**).

**Figure 6 biomedicines-11-02564-f006:**
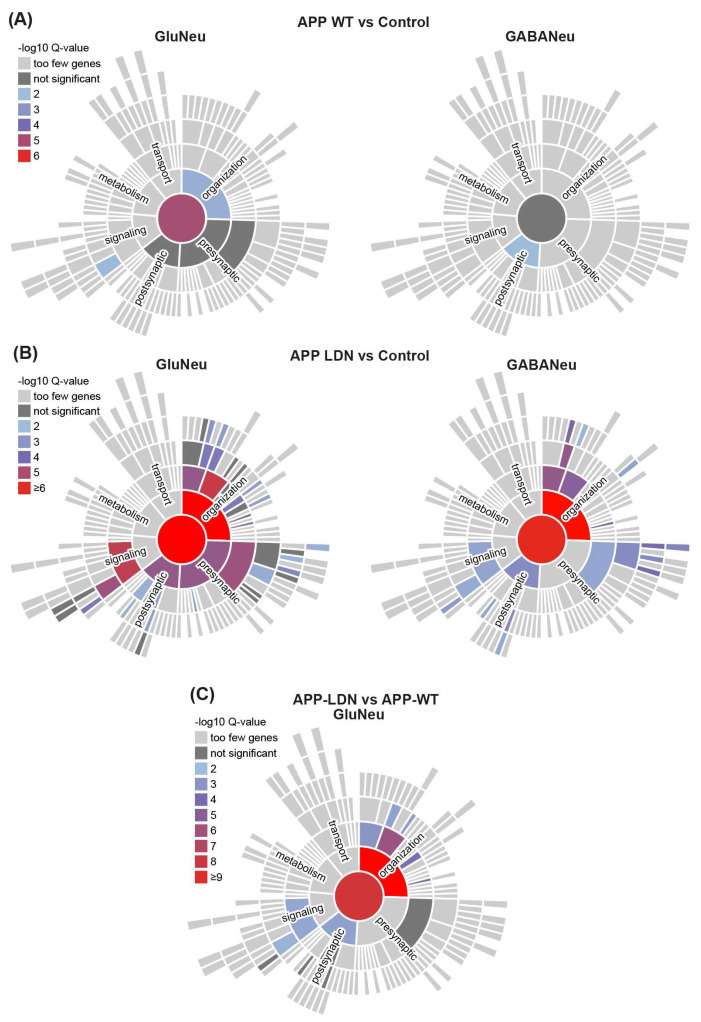
Genes modulated by CHO-APP^V717L^ conditioned medium are involved in synapse organization and signaling. (**A**–**C**) Synapse-related gene ontologies for biological processes enriched in DEGs identified in glutamatergic or GABAergic neurons when comparing CHO-APP^WT^ vs. control (**A**) or CHO-APP^V717L^ vs. control (**B**) and CHO-APP^V717L^ vs. CHO-APP^WT^ (**C**).

**Figure 7 biomedicines-11-02564-f007:**
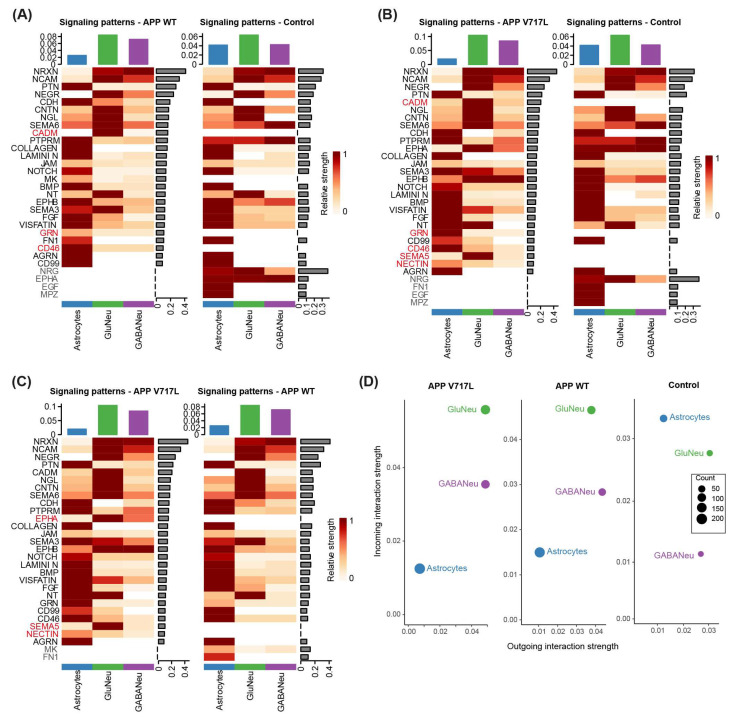
Aβ peptides modulate specific signaling pathways in hiNs/hiAs. (**A**–**C**) Heatmaps showing the relative strength of signaling pathways in astrocytes, glutamatergic and GABAergic neurons exposed to conditioned media or control treatments. Signaling pathways specific to one condition are highlighted in gray and red. (**D**) Dot plots showing the average interaction strength for incoming and outgoing signaling in astrocytes, glutamatergic and GABAergic neurons exposed to conditioned media or control treatments. Dot sizes indicate the number of ligands–receptor pairs (“counts”) identified in each cell type.

## Data Availability

The count table and metadata from snRNA sequencing experiments are available at Mendeley Data, V1 [52].

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
