# Peer review of "Amyloid-Beta Peptides Trigger Premature Functional and Gene Expression Alterations in Human-Induced Neurons"

_biomedicines, 2023, doi:10.3390/biomedicines11092564_

Round 1
Reviewer 1 Report
Comments:
The work is very interesting. The authors report results of in vitro experiments on the biochemical pathways and molecular mechanisms that mediate Aβ-mediated neuronal hyperexcitability in AD. The results allow us to advance our knowledge of how Aβ peptide deposits lead to neuronal dysfunctions.
Minors:
1.- Please unify criteria: 24 hours line 109, 24h line 24h, an hour line 131, and 24 hours line 144, etc.
2.-Please unify criteria (10μg/mL) line 108
3.- Please correct concentration 500 ul/well line 138
4.- The indication in the text for Figure 1A is missing. I suppose it is the one that appears on line 262 as Figure 1.
5.- The font size of Figure 1A and the image quality of Figures 1A and 1B should be increased.
Majors:
1.- The Results Section is very extensive from my point of view. The authors include in results, parts that are specific to the material and methods and/or discussion. Therefore, the Methods, Results and Discussion Sections should be restructured.
2.- In Figure 4 only subdivision A appears, the rest of the subdivisions are missing. Furthermore, Figure 4A is not indicated in the text. The subdivision and their indication in the text would help the understanding of the results explained in subsection 3.2.
The subdivision and indication in the text would help the understanding of the results explained in subsection 3.2.
3.- The indication in the text for Figure 2A and 2B are missing.
4.- The representation of significance is not clear in Figure 3E.
5.- The indication in the text for Figure 5B are missing. I suppose that in line 388 the figure number is missing, that is, Figure 5A-B.
In general, the style in which the results are presented could be greatly improved, for example, by restructuring each of the main sections of the paper, focusing and clearly exposing the results achieved. The results as they have been written and presented in the figures do not help with their understanding and tarnish the results found with the work.

Author Response
The work is very interesting. The authors report results of in vitro experiments on the biochemical pathways and molecular mechanisms that mediate Aβ-mediated neuronal hyperexcitability in AD. The results allow us to advance our knowledge of how Aβ peptide deposits lead to neuronal dysfunctions.
We thank the reviewer for this positive evaluation of our work.
Minors:
1.- Please unify criteria: 24 hours line 109, 24h line 24h, an hour line 131, and 24 hours line 144, etc.
2.-Please unify criteria (10μg/mL) line 108
3.- Please correct concentration 500 ul/well line 138
4.- The indication in the text for Figure 1A is missing. I suppose it is the one that appears on line 262 as Figure 1.
Thanks for highlighting these inconsistencies. We have now unified criteria and corrected text.
5.- The font size of Figure 1A and the image quality of Figures 1A and 1B should be increased.
We have updated Figure 1.
Majors:
1.- The Results Section is very extensive from my point of view. The authors include in results, parts that are specific to the material and methods and/or discussion. Therefore, the Methods, Results and Discussion Sections should be restructured.
We moved some parts of the Results to the Methods and Discussion sections, as suggested by the reviewer.
2.- In Figure 4 only subdivision A appears, the rest of the subdivisions are missing. Furthermore, Figure 4A is not indicated in the text. The subdivision and their indication in the text would help the understanding of the results explained in subsection 3.2.
We thank the reviewer for this suggestion. We have now included subdivisions B and C and their citations in the text.
3.- The indication in the text for Figure 2A and 2B are missing.
Corrected (line 283)
4.- The representation of significance is not clear in Figure 3E.
We have improved the representation of significance in Figure 3E.
5.- The indication in the text for Figure 5B are missing. I suppose that in line 388 the figure number is missing, that is, Figure 5A-B.
Thanks for pointing out this missing indication. We have corrected it.
In general, the style in which the results are presented could be greatly improved, for example, by restructuring each of the main sections of the paper, focusing and clearly exposing the results achieved. The results as they have been written and presented in the figures do not help with their understanding and tarnish the results found with the work.
We have edited the text to improve the presentation of our results.
Reviewer 2 Report
The study of Melo de Farias aims to analyze the changes in transcriptome in single human iPS neurogenic cells affected by low concentration of cell-secreted or pure beta-amyloid and thus modeling Alzheimer’s disease. The work is distinguished by thoroughness of experiments and interesting results, particularly the fact that even low concentrations of amyloid are able to change synapse-related gene expression in different types of neurons.
My comments are following.
--When studying the m/s many questions are coming mainly relating to author’s interpretation of the data. One of such concerns is why the authors think that the up-regulation of a gene expression does reflect the change in patterns of molecular signaling in amyloid-treated cells; isn’t signal transduction related to the real protein-protein interactions instead? I hope the authors are able to critically revise the discussion of their data concerning glutamatergic cells-astrocyte reactions in the AD model; the phrase about that “conditioned media treatment reduced the outgoing and incoming interaction strength in astrocytes” (lines 450-451) looks deceptive. I would also correct the fragment of the discussion on the effect of amyloid on calcium signaling (lines 514-522);
--It is rather difficult to distinguish data for the control and other cell groups. For instance, on Fig.3 does “control” (3D) mean “blank” (3E) or anything else? Generally, it is difficult to follow the authors’ principles of pairing control and amyloid-treated cells as for instance demonstrated on Fig.5-7. Can the authors give any other type of data presentation without “vs control” or “vs WT”?
--When analyzing the data for untreated (control?), WT- and V717L media-treated cells one comes to the conclusion that the factor mostly influencing transcriptome and cell type distribution (at least in astrocytes) is the medium from CHO cells; this is especially typical of data depicted on Fig.3 and 7. So, does it mean that the effect of beta-amyloid may be exaggerated?
--The list of possible ligands of beta-amyloid includes preferentially synapse proteins, though the peptide’s interactors are well established (see for instance Lazarev et al., Pharmaceuticals 2023;16(2):312. doi: 10.3390/ph16020312). On my mind this again shows that the authors prefer to look on signaling patterns in AD-mimicking cells taking in account mRNA quantities rather than real intermolecular reactions.
Author Response
The study of Melo de Farias aims to analyze the changes in transcriptome in single human iPS neurogenic cells affected by low concentration of cell-secreted or pure beta-amyloid and thus modeling Alzheimer’s disease. The work is distinguished by thoroughness of experiments and interesting results, particularly the fact that even low concentrations of amyloid are able to change synapse-related gene expression in different types of neurons. 
We thank the reviewer for recognizing the significance of our results.
My comments are following.
--When studying the m/s many questions are coming mainly relating to author’s interpretation of the data. One of such concerns is why the authors think that the up-regulation of a gene expression does reflect the change in patterns of molecular signaling in amyloid-treated cells; isn’t signal transduction related to the real protein-protein interactions instead? I hope the authors are able to critically revise the discussion of their data concerning glutamatergic cells-astrocyte reactions in the AD model; the phrase about that “conditioned media treatment reduced the outgoing and incoming interaction strength in astrocytes” (lines 450-451) looks deceptive.
We thank the reviewer for these questions. In fact, there are different analyses described in the MS and we may have failed to clearly discuss our findings. In the first moment, we performed classical analysis of differential gene expression in each cell type identified in our cultures (NPCs, Astrocytes, Glutamatergic neurons and GABAergic neurons). In this analysis, we show that treatment with conditioned medium containing higher concentrations of amyloid-beta peptides is sufficient to elicit changes in gene expression in astrocytes and neurons (Figure 4). Moreover, our comparison between APP V717L and APP WT groups show reveals a predominant change in gene expression among glutamatergic neurons, which suggests a particular effect of Aβ peptides in these cells (see Figure 1).
Next, we identified potential signaling pathways among astrocytes and neurons based on the expression of ligand-receptor pairs (Figure 7). This is different from differential gene expression analysis performed before and includes different statistics to calculate the probability and strength of intercellular communication patterns. In this case, signaling patterns are calculated for each condition and can be then compared side by side, as shown in Figure 7. The panel 7D, for example, indicates the relative interaction strengths including ligands (outgoing) and receptors (incoming) expressed in the different cell types. The shift in the position of astrocytes and neurons in the “control” and “APP” conditions is the source of the sentence “conditioned media treatment reduced the outgoing and incoming interaction strength in astrocytes” (lines 450-451). In the new version of the manuscript, we tried to explain it in more details.
I would also correct the fragment of the discussion on the effect of amyloid on calcium signaling (lines 514-522); 
We thank the reviewer for this suggestion. The sentence was modified as follows:
“The alterations in neuronal calcium dynamics together with the gene expression pattern of known Aβ1-42 ligands observed in glutamatergic and GABAergic neurons in our cultures may suggest a direct effect of this peptide in the regulation of neuronal functional properties.”
--It is rather difficult to distinguish data for the control and other cell groups. For instance, on Fig.3 does “control” (3D) mean “blank” (3E) or anything else? Generally, it is difficult to follow the authors’ principles of pairing control and amyloid-treated cells as for instance demonstrated on Fig.5-7. Can the authors give any other type of data presentation without “vs control” or “vs WT”?   
We apologize for this confusion. In fact, there is only one control, which is the non-conditioned/blank medium. We have changed the labels in the figures to avoid further confusions.
Regarding the presentation of data in Figures 5-7, we can only identify differentially expressed genes (and the GO terms enriched for them) when comparing two conditions. Therefore, we cannot change the way to analyze/present the data.
--When analyzing the data for untreated (control?), WT- and V717L media-treated cells one comes to the conclusion that the factor mostly influencing transcriptome and cell type distribution (at least in astrocytes) is the medium from CHO cells; this is especially typical of data depicted on Fig.3 and 7. So, does it mean that the effect of beta-amyloid may be exaggerated? 
We thank the reviewer for raising this important point. In fact, we find a slightly higher proportion of astrocytes in control compared to other conditions (Figure 3E). However, and this is the power of single-cell RNA sequencing, our analyses of differential gene expression and intercellular communication are not influenced by that shift in cell population, since all statistical tests used for DSEq2 and CellChatDB are performed independently for astrocytes, glutamatergic neurons or GABAergic neurons.
--The list of possible ligands of beta-amyloid includes preferentially synapse proteins, though the peptide’s interactors are well established (see for instance Lazarev et al., Pharmaceuticals 2023;16(2):312. doi: 10.3390/ph16020312). On my mind this again shows that the authors prefer to look on signaling patterns in AD-mimicking cells taking in account mRNA quantities rather than real intermolecular reactions. 
We are not sure to fully understand this comment, but we believe that the reviewer is talking about the expression of known Aβ ligands described in Figure 3F. As described in the text, this list was curated from the literature and is reviewed by Chen et al. 2017 (doi.org/10.1038/aps.2017.28). Although this is not an exhaustive list, it includes most receptors known to mediate several cellular effects of Aβ in neural cells.
In the reference cited by the reviewer, the authors discuss other potential interactors, such as UBA1, GFAP, HSPA1A (HSP70), PTGFRN, GAPDH, TTR, CST3, KAT5 (TIP60), FBLN1, SLC25A4 and EME1. We checked the expression of these genes in our cells and the results are attached.
Lastly, we agree that studying intermolecular interactions in the protein level is an important step towards a more definitive description of the molecular pathways mediated by Aβ in neural cell. However, we also believe that data generated from single-cell analyses, where the expression of a putative interactor can be evaluated in a cell type-specific manner also contribute to interpret the results from intermolecular reactions (if protein A is expressed in neurons, we can expect that the interaction will happen in these cells and not in astrocytes, for example).

Round 2
Reviewer 1 Report
The authors have taken into account the suggestions provided.
The version of the paper provided has improved in quality.